# Contributions of neighborhood social environment and air pollution exposure to Black-White disparities in epigenetic aging

Isabel Yannatos [1], Shana Stites [2], Rebecca T. Brown[3,4,5,6], Corey T. McMillan[1]*

1 Department of Neurology, University of Pennsylvania, Philadelphia, Pennsylvania, United States of America, 2 Department of Psychiatry, Perelman School of Medicine, University of Pennsylvania, Philadelphia, Pennsylvania, United States of America, 3 Division of Geriatric Medicine, Perelman School of Medicine at the University of Pennsylvania, Philadelphia, Pennsylvania, United States of America, 4 Geriatrics and Extended Care, Corporal Michael J. Crescenz Veterans Affairs Medical Center, Philadelphia, Pennsylvania, United States of America, 5 Center for Health Equity Research and Promotion, Corporal Michael J. Crescenz Veterans Affairs Medical Center, Philadelphia, Pennsylvania, United States of America, 6 Leonard Davis Institute of Health Economics, University of Pennsylvania, Philadelphia, Pennsylvania, United States of America

* cmcmilla@pennmedicine.upenn.edu

**Data Availability Statement:** Data Availability: Data used in this study are available through a third-party, the Health and Retirement Study. However, once a researcher has obtained access to the data,

## Abstract

Racial disparities in many aging-related health outcomes are persistent and pervasive among older Americans, reflecting accelerated biological aging for Black Americans compared to White, known as weathering. Environmental determinants that contribute to weathering are poorly understood. Having a higher biological age, measured by DNA methylation (DNAm), than chronological age is robustly associated with worse age-related outcomes and higher social adversity. We hypothesize that individual socioeconomic status (SES), neighborhood social environment, and air pollution exposures contribute to racial disparities in DNAm aging according to GrimAge and Dunedin Pace of Aging methylation (DPoAm). We perform retrospective cross-sectional analyses among 2,960 non-Hispanic participants (82% White, 18% Black) in the Health and Retirement Study whose 2016 DNAm age is linked to survey responses and geographic data. DNAm aging is defined as the residual after regressing DNAm age on chronological age. We observe Black individuals have significantly accelerated DNAm aging on average compared to White individuals according to GrimAge (239%) and DPoAm (238%). We implement multivariable linear regression models and threefold decomposition to identify exposures that contribute to this disparity. Exposure measures include individual-level SES, census-tract-level socioeconomic deprivation and air pollution (fine particulate matter, nitrogen dioxide, and ozone), and perceived neighborhood social and physical disorder. Race and gender are included as covariates. Regression and decomposition results show that individual-level SES is strongly associated with and accounts for a large portion of the disparity in both GrimAge and DPoAm aging. Higher neighborhood deprivation for Black participants significantly contributes to the disparity in GrimAge aging. Black participants are more vulnerable to fine particulate matter exposure for DPoAm, perhaps due to individual- and neighborhood-level SES, which may contribute to the disparity in DPoAm aging. DNAm aging may play a role in the environment "getting

they can use our code available at github.com/pennbindlab to recreate the minimal data set. Description of the data set and source: The Health and Retirement Study (HRS) is sponsored by the National Institute on Aging (grant number NIA U01 AG009740) and is conducted by the University of Michigan. It is a longitudinal panel survey study that collects in-depth survey interviews and biological samples from a representative sample of Americans 50 and older. Study participants' geographic location and Contextual Data Resource data are available only under special agreement because they contain sensitive and/or confidential information. Verification of permission to use the data set: Restricted Data Agreement # 2021-084 was approved on October 15, 2021. Information to apply to gain access: Researchers must apply for access through HRS at the site https://hrs.isr.umich.edu/data-products/restricted-data. The application and use of data are free of charge. IRB approval is required. All questions related to HRS Restricted data should be sent to hrsrdaapplication@umich.edu.

**Funding:** This work was supported by National Institute on Aging: R01-AG066152 (CM), R01-AG070885 (RB), P30-AG072979 (CM). Additional support includes Pennsylvania Department of Health (2019NF4100087335; CM), and Penn Institute on Aging (CM). National Institute on Aging: https://www.nia.nih.gov Pennsylvania Department of Health: https://www.health.pa.gov/Pages/default.aspx Penn Institute on Aging: https://www.med.upenn.edu/aging/ The funders had no role in study design, data collection and analysis, decision to publish, or preparation of the manuscript.

**Competing interests:** The authors have declared that no competing interests exist.

under the skin", contributing to age-related health disparities between older Black and White Americans.

## Introduction

There are severe racial disparities in age-related health in the United States. Black Americans have earlier onset, higher prevalence, and reduced survival of age-related diseases relative to their White counterparts due to the "cumulative impact of repeated experience with social or economic adversity and political marginalization" [1, 2]. Early health deterioration has been termed weathering and reflects accelerated aging [3, 4]. Across many biomarkers of aging, Black Americans are biologically older than their White counterparts of the same chronological age [5]. There is evidence that individual socioeconomic status (SES), neighborhood deprivation and segregation, and discrimination may play a role in these disparities [6–8]. However, the social and structural determinants of weathering are not well understood and the contributions of individual and neighborhood factors know to impact age-related health have not been quantified.

Social epigenetics posits that DNA methylation (DNAm) is a mediating link between social and structural determinants of health and both age-related health outcomes and health disparities, though there is limited work directly testing this hypothesis [9, 10]. Social and structural determinants of health are the conditions in which individuals are born, live, learn, work, and age [11]. Structural factors such as racial segregation and discrimination, exclusionary economic policy, and environmental racism influence an individual's socioeconomic resources and exposure to environmental conditions, such as the social environment of one's neighborhood and physical pollutants. These determinants are known to be important for age-related health and health disparities, but there is limited understanding of the biological mechanisms by which they affect health outcomes. There is evidence that DNAm aging may be one such mechanism [10, 12].

Markers of biological aging using DNAm have emerged as robust measures of weathering, especially GrimAge and Dunedin Pace of Aging Methylation (DPoAM) [13, 14]. An advantage of these measures is they were trained to predict phenotypic age based on biomarkers and mortality rather than chronological age alone, which may more accurately reflect biological aging processes [12]. Both GrimAge and DPoAm demonstrate racial disparities, reflecting weathering, and strongly predict many age-related outcomes, including lung disease, cognitive decline, functional decline, and mortality [13–20].

Prior evidence from the Health and Retirement Study, the cohort focus of this study, suggests that the hazard ratio of mortality is 2.32 and 1.71 per standard deviation increase in GrimAge and DPoAm acceleration, respectively [5]. Acceleration in GrimAge or DPoAm was further associated with racial disparities and an increased risk of prevalent and incident functional limitations and chronic conditions and poorer self-rated health as well as age-related outcomes: GrimAge and DPoAm mediated 13–92% of disparities in functional status and decline, self-rated health, and mortality [5]. Although DNAm aging has been associated with social and structural determinants, including individual SES and aspects of the neighborhood social environment, to the best of our knowledge, prior studies have not quantified the contribution of specific determinants to the racial disparity in DNAm aging, nor examined whether GrimAge and DPoAm are associated with air pollution exposure [8, 13, 14, 19–21].

The neighborhood social and physical environments are important determinants for age-related health. Neighborhood socioeconomic deprivation, perceived neighborhood disorder

(i.e., reporting less social cohesion and safety and more physical disorder), and air pollution exposure have all been linked to health status and decline in older adults. A growing body of research shows that living in a neighborhood with a greater proportion of people with low socioeconomic resources is associated with many adverse health outcomes for older adults regardless of their individual SES [22–31]. There are mixed results on the health effects of perceived social and physical disorder in one's neighborhood, but disorder is associated with risk of functional decline, cardiovascular disease, and dementia [25, 28, 32–36]. Exposure to fine particulate matter ($PM_{2.5}$) air pollution is an established risk factor for mortality and several age-related diseases and nitrogen dioxide ($NO_2$) is associated with several of these outcomes as well, while findings for ozone ($O_3$) are mixed and show weak to no associations with health outcomes [37–41]. The risks of air pollution exposure may be even higher for older adults than for the general population [39].

Historic and current structural and environmental racism and persistent racial residential segregation result in inequitable distributions of these social determinants, where Black Americans have fewer socioeconomic resources and higher exposure to unfavorable and unhealthy conditions in their neighborhood environment [42–44]. The large impact of the inequitable distribution of individual SES on health disparities is well documented [45]. Black older adults are more likely to live in a neighborhood with higher deprivation, disorder, and $PM_{2.5}$ and $NO_2$ pollution than their White counterparts and these inequities also contribute to health disparities [33, 46–48]. For example, neighborhood socioeconomic composition explains large portions of the racial disparities in COVID-19 infection in Chicago and self-rated health among older adults [49, 50]. Neighborhood stress contributes to the racial disparity in hypertension [51]. Air pollution contributes to racial disparities in hypertension, Alzheimer's disease, and likely other age-related diseases [52–54].

Furthermore, the relationship between measured environmental exposures and age-related outcomes may be different between racial groups. Black Americans appear to have greater risk from the same amount of $PM_{2.5}$ pollution than their White counterparts; Black Medicare beneficiaries had three times greater risk of mortality and Black women had twice the risk of Alzheimer's Disease due to $PM_{2.5}$ [53, 55]. There is also some evidence that risk of functional limitations due to neighborhood physical disorder may differ between racial groups [33]. Since race is a social construct, any differences in risk between racial groups can be attributed to structural and social determinants that affect either amount of exposure or defenses against a detrimental neighborhood exposure. Research is needed to determine whether risk from social environment exposures differ, whether differences contribute to racial disparities in outcomes, and the determinants that influence these differences.

We selected an approach using threefold decomposition, a technique commonly used in econometrics and social sciences to investigate disparities between groups, because it allows us to evaluate how the both the distribution of an exposure and differences in risk contribute to disparities. This method decomposes a difference in an outcome between two groups into three components that are 1) explained by differences in the level of explanatory variables between the groups, 2) explained by differences in the effect of explanatory variables on the outcome between the groups, and 3) an unexplained portion [56]. This is accomplished by estimating group-specific regression models and using a counterfactual approach, where the change in the outcome disparity is evaluated after replacing the disadvantaged group's covariate and coefficient values with those of the reference group.

In this study we investigate the extent to which different levels of neighborhood exposures and different risks due to those exposures between Black and White older Americans contribute to racial disparities in DNAm aging. We first hypothesize that individual-level SES greatly contributes to the DNAm aging disparity but does not fully explain it. We further hypothesize

that neighborhood exposures with a larger racial disparity and stronger association with biological aging contribute more to the DNAm aging disparity (specifically, neighborhood deprivation, $PM_{2.5}$, and $NO_2$ contribute more than neighborhood disorder and $O_3$, which is included as a negative control). Finally, we hypothesize that Black participants have greater risk due to $PM_{2.5}$ than White and perhaps to other neighborhood exposures as well. We test these hypotheses using a large nationally representative sample of adults 50 and over, two robust markers of biological aging as outcomes, and two complementary analytical techniques, regression and decomposition. This study builds on the weathering literature by identifying specific environmental-level factors that contribute to racial disparities in biological aging, including air pollution, and by examining not only different levels of environmental exposures between racial groups but also potentially different levels of risk.

## Methods

### Data

The Health and Retirement Study (HRS) is sponsored by the National Institute on Aging (grant number NIA U01 AG009740) and is conducted by the University of Michigan. Surveys are administered by phone or in person biannually with a nationally representative population of Americans aged 50 or older [57]. In 2016, a subset of HRS participants provided a venous blood sample of which a representative subsample (N = 4018) was selected for DNA methylation (DNAm) measurement [58]. We link epigenetic data with data from the 2016 wave of the HRS survey, the Psychosocial and Lifestyle Questionnaire (2008–2014), and the HRS Contextual Data Resource (CDR). The Psychosocial and Lifestyle Questionnaire is given to half of HRS respondents in each wave as a self-administered questionnaire after completing a face-to-face interview [59]. Response rates are 73–83%. The CDR is a restricted data set which includes geographic identifiers and data drawn from sources such as the American Community Survey and Environmental Protection Agency (EPA) [60, 61].

### Ethics statement

All data were collected by HRS following written informed consent under a protocol approved by an IRB at the University of Michigan. We report secondary analyses approved by an IRB at the University of Pennsylvania.

### Population

We include self-identified non-Hispanic White and Black individuals in the DNAm sub-sample with complete exposure and covariate data. 366 individuals were excluded for missing any variables. Our final sample comprises 2,960 participants; 82% (N = 2438) are White, 18% (N = 522) are Black.

### Measures

**DNA methylation aging.** Whole blood samples collected in EDTA tubes were sent to the CLIA-certified Advanced Research and Diagnostic Laboratory at the University of Minnesota for centralized processing [58]. DNAm was measured at 866,091 CpG sites using the Infinium Methylation EPIC BeadChip. Samples were randomized across plates, run in duplicate, and quality controlled. Values for GrimAge and Dunedin Pace of Aging methylation (DPoAm) were estimated based on published CpG sites and weights [13, 14]. GrimAge was trained using data from the Framingham Heart Study Offspring Cohort. Elastic net regression was used to create DNAm proxies for plasma protein biomarkers and smoking pack years, then the

DNAm proxies were combined to predict time to death. The resulting GrimAge value is transformed to units of years. DPoAm was trained using data from the birth cohort Dunedin Study. The rate of change in 18 biomarkers from ages 26 to 38 were combined in a Pace of Aging measure, then elastic net regression was used to select CpG sites that predict Pace of Aging. DPoAm is reported in units of biological years per chronological year as a measure of pace of aging. For both, we regress value on chronological age and use the residual to measure individuals' DNAm aging, consistent with previous work [62]. A residual greater than zero indicates higher clock value than expected based on chronological age, i.e., accelerated DNAm aging. We repeat this regression including ten ancestry-informative principal components, as described below, and without population weighting and report results in S1 Table. We then divide by the root mean square to scale the residuals for ease of comparison between the two clock measures.

**Environmental exposures.** For exposure to neighborhood socioeconomic deprivation and air pollution, we use census-tract-level data based on participants' census tract of residence in 2014, the most recent year prior to the outcome measurement.

*Social Deprivation Index (SDI)*. The SDI is a composite measure of socioeconomic deprivation based on seven characteristics, such as percent of the population under the poverty line or unemployed, constructed using data from the American Community Survey [63, 64]. We use the publicly available census-tract-level 2015 SDI score.

*Air pollution*. Mean level of fine particulate matter air pollution ($PM_{2.5}$), Nitrogen Dioxide ($NO_2$), and Ozone ($O_3$) per census tract are available from the EPA as part of the CDR. We use average 2014 levels at participants' 2014 residential census tracts for $PM_{2.5}$ and $O_3$. The most recent data available for $NO_2$ are for 2010, so we use the average 2010 level at participants' 2010 residential census tract.

*Neighborhood disorder*. We use participant evaluations of their neighborhood in the Psychosocial and Lifestyle Questionnaire as measures of perceived neighborhood social and physical disorder [59]. Respondents rated their agreement on a seven-point scale to each of eight statements. Four statements relate to social disorder (sense of belonging, trustworthiness, friendliness, and helpfulness among neighbors) and four to physical disorder (presence of vandalism, litter, vacant buildings, and sense of safety) in their neighborhood (defined as everywhere within a 20-minute walk or half a mile of your home). We use the most recent response from 2008–2014 for each respondent and the average score across the four items, log-transformed and scaled to normalize distributions. Higher scores indicate greater perceived disorder.

**Covariates.** We use self-reported race, ethnicity, and gender. Participants who identified as Hispanic or Latino or as a race other than Black or White are excluded. The HRS survey treats gender as binary and does not distinguish between assigned sex or gender identity. "Sex-mismatched" blood samples were removed from DNAm measurement, so there is likely no representation of trans or intersex individuals in this dataset. Education is categorized based on years of formal schooling completed. We use an index of household income and wealth to avoid collinearity. Household income is the sum of respondent and spouse annual income. Household wealth is the sum of all assets, including second homes, minus debts. We log transform income and wealth, calculate a Z score for each, average the Z scores, then use weighted quartiles. Race, gender, education, and quartile of wealth/income are included as covariates in all analyses. We do not include chronological age as a covariate because it has a null association with DNAm aging (coefficient of zero) in all models. Including chronological age has no effect on any results in regression models.

**Sample weights.** HRS provides sample weights for the DNAm subsample to adjust for probability of participation. More detail on weights can be found in the documentation [58].

We impute missing sample weights (N = 143 missing) using the mean value and include weighting in all regression analyses.

**Ancestry-informative principal components.** We perform a supplementary analysis correcting for ancestry-informative genetic markers to account for population stratification, which varies between ancestry groups and can influence DNAm [65]. Ten ancestry-specific principal components are available for a portion of the White (European ancestry) and Black (African ancestry) participants and were included as covariates in supplemental regression of GrimAge and DPoAm values on chronological age (S1 Table) [66].

### Analytic approach

All analyses are conducted in R statistical software (version 4.2.1) and code is available at github.com/pennbindlab [67]. Models are informed by a directed acyclic graph showing proposed relationships between all measures (Fig 1). Each neighborhood exposure is treated independently, represented by "Neighborhood" in the figure. Race influences education, income, wealth, neighborhood exposures, and DNAm aging due to various manifestations of structural racism. Age and gender are important confounders of these effects. We use two complementary analytic strategies.

**Linear regression models.** We first implement stepwise linear regression. We perform univariate linear regression to assess the association between DNAm aging and race alone, which defines the total racial disparity. We then add individual-level covariates (gender, education, and wealth/income). To evaluate associations between DNAm aging and the environment we then add each environmental exposure individually (SDI, perceived social disorder, perceived physical disorder, $PM_{2.5}$, $NO_2$, and $O_3$). Variance inflation factors for all models

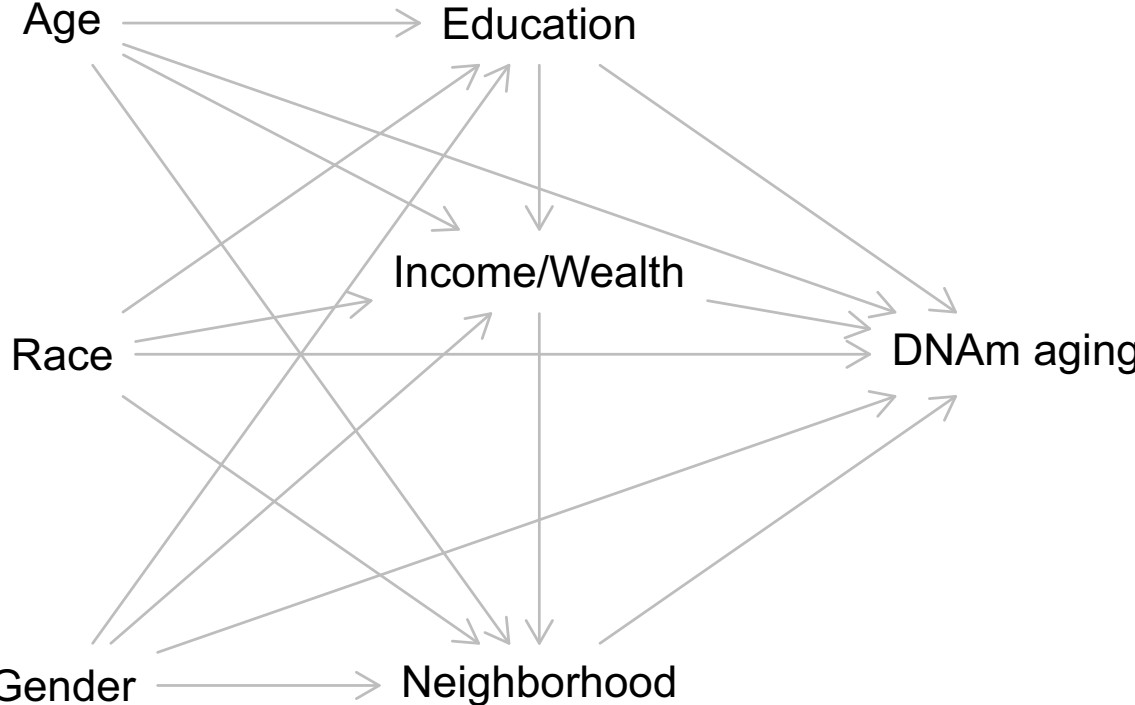

**Fig 1. Directed acyclic graph showing relationships between variables.** Race, age, gender, education, income/wealth, and neighborhood exposures all have direct effects on DNAm aging as well as indirect effects through the pathways indicated. Race, age, and gender are associated with education, income, and wealth, all of which influence one's residential location and neighborhood exposures.

are $\leq 1.5$, indicating low multicollinearity. We perform sensitivity analysis to assess whether residential mobility biases the results by excluding individuals who moved between 2010 and 2016 (N = 667) (S2 and S3 Tables). We assess whether the relationships between the exposures and DNAm aging differs between racial groups by adding interaction terms with race. We assess factors that may influence vulnerability to $PM_{2.5}$ exposure by adding interaction terms between $PM_{2.5}$ and gender, education, wealth/income, and neighborhood social environment measures (SDI, perceived social and physical disorder). All models include sample weighting, as described above. P-values are Bonferroni corrected to account for six comparisons. AIC (Aikake's Information Criterion) is shown as a goodness-of-fit indicator.

**Threefold decomposition.** Next, we implement Kitagawa-Blinder-Oaxaca decomposition to quantify how each covariate and environmental measure contribute to the disparity in DNAm aging using the package Oaxaca() (version 0.1.5) [68, 69]. Threefold decomposition separates these contributions into endowment, coefficient, and interaction effects. The endowment effect indicates the portion of the disparity due to differences in the levels of the explanatory variables between groups. The coefficient effect reflects the portion of the disparity due to differences in the coefficient of the variables between groups, i.e. different associations or effects of the exposures between groups. The interaction effect accounts for differences in endowments and coefficients that exist simultaneously. We use linear regression as the underlying models for decomposition and run 10,000 bootstrap samples to calculate 95% confidence intervals.

## Results

### There are large racial disparities in DNA methylation aging and environmental exposures

Descriptive statistics by race are shown in Table 1. The overall sample has an average age of 71 years and is 41% male. Black participants are significantly younger, more likely to be women, less educated, and have lower income/wealth than White participants on average (p<0.001). They also have higher levels of all environmental exposures except ozone ($O_3$); Black participants are exposed to higher levels of neighborhood social deprivation, perceived social and physical disorder, fine particulate matter ($PM_{2.5}$), and nitrogen dioxide ($NO_2$) (p<0.001). Effect sizes for these disparities are medium to large (Cohen's d > 0.5), with the difference in social deprivation being particularly striking. Black participants on average have higher rates of DNA methylation (DNAm) aging than White participants in both GrimAge and Dunedin Pace of Aging methylation (DPoAm) aging (p <0.001); Black participants on average have accelerated DNAm aging while White participants have slightly decelerated DNAm aging. Scaled residuals are shown in Table 1 and used in all analyses for ease of comparison between the two DNAm aging measures. Unscaled residuals show that the difference between Black and White participants' DNAm aging is 1.4 biological years according to GrimAge, and 0.03 biological years (or 0.36 biological months) per chronological year according to DPoAm (S1 Table). The effect size for disparities in all DNAm aging measures are moderate (Cohen's d ~ 0.3).

Sensitivity analyses show that the absolute disparity in DNAm aging remains the same whether sample weights are applied and whether ancestry-informative genetic markers are included in the regression (S1 Table). The racial disparity in DNAm aging is not due to sampling bias nor genetic background.

### Stepwise multivariable regression models

Regressing DNAm aging on race gives the total disparity between White and Black participants in GrimAge and DPoAm aging in scaled units, shown in the first column of Tables 2 and 3,

**Table 1. Sample characteristics and racial disparities.**

| Characteristic | Overall, N = 2,960[1] | White, N = 2,438[1] | Black, N = 522[1] | p-value[2] | Effect Size[3] |
|---|---|---|---|---|---|
| **Age (years)** | 71.33 (9.53) | 72.08 (9.54) | 67.81 (8.69) | <0.001 | 0.468 |
| **Gender** | | | | <0.001 | 0.085 |
| Male | 1,218 (41%) | 1,051 (43%) | 167 (32%) | | |
| Female | 1,742 (59%) | 1,387 (57%) | 355 (68%) | | |
| **GrimAge aging** | 0.03 (1.00) | -0.02 (0.99) | 0.28 (1.03) | <0.001 | 0.304 |
| **DunedinPoAm aging** | 0.03 (1.00) | -0.03 (0.98) | 0.31 (1.04) | <0.001 | 0.334 |
| **Education** | | | | <0.001 | 0.164 |
| College + | 839 (28%) | 740 (30%) | 99 (19%) | | |
| Some College | 773 (26%) | 644 (26%) | 129 (25%) | | |
| High School | 974 (33%) | 803 (33%) | 171 (33%) | | |
| < High School | 374 (13%) | 251 (10%) | 123 (24%) | | |
| **Wealth/Income Quartile** | | | | <0.001 | 0.325 |
| 4 | 702 (24%) | 680 (28%) | 22 (4.2%) | | |
| 3 | 751 (25%) | 671 (28%) | 80 (15%) | | |
| 2 | 803 (27%) | 644 (26%) | 159 (30%) | | |
| 1 | 704 (24%) | 443 (18%) | 261 (50%) | | |
| **Social Deprivation Index** | -0.12 (0.98) | -0.32 (0.89) | 0.79 (0.80) | <0.001 | 1.31 |
| **Social Disorder** | -0.06 (0.96) | -0.16 (0.93) | 0.41 (0.98) | <0.001 | 0.600 |
| **Physical Disorder** | -0.06 (0.95) | -0.17 (0.92) | 0.46 (0.96) | <0.001 | 0.670 |
| **$PM_{2.5}$ (µg/m³)** | 9.52 (1.91) | 9.36 (1.94) | 10.26 (1.61) | <0.001 | 0.506 |
| **Ozone (µg/m³)** | 38.17 (3.89) | 38.40 (4.01) | 37.09 (3.04) | <0.001 | 0.367 |
| **$NO_2$ (ppb)** | 8.23 (4.14) | 7.78 (3.80) | 10.37 (4.93) | <0.001 | 0.590 |

Sample characteristics overall and by race. Significance of comparison between White and Black groups shown.

[1]Mean (SD); n (%)

[2]Wilcoxon rank sum test; Pearson's Chi-squared test

[3]Cohen's D; Cramer's V

respectively. The disparity in GrimAge is 0.30 units (95% CI 0.18, 0.42; p<0.001), and the disparity in DPoAm is 0.35 units (95% CI 0.23, 0.47; p<0.001). Adding individual-level covariates in the second column shows that gender and individual-level socioeconomic status (SES) are significantly associated with both GrimAge and DpoAm aging. Being female (GrimAge β = -0.73, 95% CI -0.8, -0.67; DpoAm β = -0.2, 95% CI -0.27, -0.13; p< 0.001) is associated with decreased DNAm aging while having less education (GrimAge β = 0.51, 95% CI 0.38, 0.63; DPoAm β = 0.39, 95% CI 0.26, 0.53; p<0.001 for less than high school compared to college or more), or having lower wealth/income (GrimAge β = 0.53, 95% CI 0.43, 0.64; DPoAm β = 0.32, 95% CI 0.21, 0.44; p<0.001 for the lowest quartile compared to the highest quartile) is associated with increased DNAm aging. These associations tend to be larger in magnitude with GrimAge than DPoAm aging. Adding these individual-level factors reduces the association between GrimAge aging and race from 0.30 to 0.13 (95% CI 0.02, 0.24) units and to non-significance, implying that these factors strongly contribute to the racial disparity in GrimAge aging. The association between DPoAm aging and race remains significant after adding these factors, but the magnitude is reduced from 0.35 to 0.21 (95% CI 0.09, 0.33, p<0.01) units, implying that individual-level SES contributes somewhat to the disparity in DPoAm aging.

We then add each environmental exposure separately to the model. The Social Deprivation Index (SDI) is significantly associated with GrimAge aging (β = 0.06, 95% CI 0.02, 0.09; p<0.05), and further reduces the association between GrimAge aging and race to 0.08 (95% CI

**Table 2. GrimAge aging: Multivariable associations with individual covariates and neighborhood exposures.**

| GrimAge[1] | Total disparity[1] | Individual SES[1] | SDI[1] | Social Disorder[1] | Physical Disorder[1] | PM$_{2.5}$[1] | Ozone[1] | NO$_2$[1] |
|---|---|---|---|---|---|---|---|---|
| **Race** | | | | | | | | |
| White | — | — | — | — | — | — | — | — |
| Black | 0.30*** (0.18,0.42) | 0.13 (0.02,0.24) | 0.08 (-0.03,0.20) | 0.11 (0.00,0.22) | 0.12 (0.01,0.23) | 0.13 (0.02,0.24) | 0.13 (0.02,0.24) | 0.13 (0.02,0.24) |
| **Gender** | | | | | | | | |
| Male | | — | — | — | — | — | — | — |
| Female | | -0.73*** (-0.80,-0.67) | -0.73*** (-0.79,-0.66) | -0.72*** (-0.79,-0.66) | -0.73*** (-0.79,-0.66) | -0.73*** (-0.80,-0.67) | -0.73*** (-0.80,-0.67) | -0.73*** (-0.80,-0.67) |
| **Education** | | | | | | | | |
| College + | | — | — | — | — | — | — | — |
| Some College | | 0.26*** (0.17,0.34) | 0.25*** (0.17,0.34) | 0.25*** (0.17,0.34) | 0.25*** (0.17,0.34) | 0.26*** (0.17,0.34) | 0.26*** (0.17,0.34) | 0.26*** (0.17,0.34) |
| High School | | 0.30*** (0.21,0.38) | 0.29*** (0.20,0.38) | 0.30*** (0.21,0.38) | 0.29*** (0.21,0.38) | 0.29*** (0.21,0.38) | 0.30*** (0.21,0.38) | 0.30*** (0.21,0.38) |
| < High School | | 0.51*** (0.38,0.63) | 0.50*** (0.37,0.62) | 0.50*** (0.38,0.63) | 0.50*** (0.38,0.63) | 0.51*** (0.38,0.63) | 0.51*** (0.38,0.63) | 0.51*** (0.38,0.63) |
| **Quartile Wealth/ Income** | | | | | | | | |
| 4 | | — | — | — | — | — | — | — |
| 3 | | 0.15** (0.06,0.24) | 0.14* (0.05,0.23) | 0.15** (0.06,0.23) | 0.15** (0.06,0.24) | 0.15** (0.06,0.24) | 0.15** (0.06,0.24) | 0.15** (0.06,0.24) |
| 2 | | 0.41*** (0.31,0.50) | 0.38*** (0.28,0.47) | 0.40*** (0.30,0.49) | 0.40*** (0.31,0.50) | 0.41*** (0.31,0.50) | 0.41*** (0.31,0.50) | 0.41*** (0.31,0.50) |
| 1 | | 0.53*** (0.43,0.64) | 0.49*** (0.38,0.60) | 0.52*** (0.41,0.62) | 0.52*** (0.42,0.63) | 0.53*** (0.43,0.64) | 0.53*** (0.43,0.64) | 0.53*** (0.43,0.64) |
| **Neighborhood Exposure** | | | 0.06* (0.02,0.09) | 0.04 (0.00,0.07) | 0.02 (-0.02,0.06) | 0.00 (-0.01,0.02) | 0.00 (-0.01,0.01) | 0.00 (-0.01,0.01) |
| (Intercept) | -0.05* (-0.09,-0.02) | -0.09 (-0.17,-0.02) | -0.05 (-0.13,0.03) | -0.08 (-0.16,-0.01) | -0.08 (-0.16,-0.01) | -0.12 (-0.29,0.06) | -0.15 (-0.46,0.17) | -0.10 (-0.20,0.01) |
| R$^2$ | 0.008 | 0.213 | 0.215 | 0.214 | 0.213 | 0.213 | 0.213 | 0.213 |
| AIC | 9,064 | 8,393 | 8,387 | 8,390 | 8,394 | 8,395 | 8,395 | 8,395 |

Results of linear regression models with GrimAge aging as the outcome.

[1]ß (95% confidence interval)

*p<0.05

**p<0.01

***p<0.001

-0.03, 0.2) units (Table 2, column 3). The SDI is not significantly associated with DPoAm aging after correcting for multiple comparisons but has a positive coefficient (β = 0.04, 95% CI 0.0, 0.08) (Table 3, column 3). Adding SDI reduces the magnitude and significance of the association between DPoAm aging and race to 0.18 (95% CI 0.05, 0.18; p<0.05) units. Together these results imply that lower neighborhood socioeconomic resources are associated with higher GrimAge aging and contribute to the Black-White disparity in DNAm aging.

The association between perceived social disorder and GrimAge aging has a positive coefficient (β = 0.04, 95% CI 0.0, 0.07) but is not significant after correcting for multiple comparisons (Table 2, column 4). Adding social disorder has a smaller effect on the association with race than adding SDI but reduces the association slightly to 0.11 units (95% CI 0.0, 0.22). Perceived social disorder did not have an association nor affect the racial disparity in DPoAm

**Table 3. DPoAm aging: Multivariable associations with individual covariates and neighborhood exposures.**

| DPoAm[1] | Total disparity[1] | Individual SES[1] | SDI[1] | Social Disorder[1] | Physical Disorder[1] | PM$_{2.5}$[1] | Ozone[1] | NO$_2$[1] |
|---|---|---|---|---|---|---|---|---|
| **Race** | | | | | | | | |
| White | — | — | — | — | — | — | — | — |
| Black | 0.35*** (0.23,0.47) | 0.21** (0.09,0.33) | 0.18* (0.05,0.30) | 0.20** (0.08,0.32) | 0.21** (0.09,0.33) | 0.21** (0.09,0.33) | 0.21** (0.09,0.33) | 0.20** (0.08,0.32) |
| **Gender** | | | | | | | | |
| Male | | — | — | — | — | — | — | — |
| Female | | -0.20*** (-0.27,-0.13) | -0.20*** (-0.27,-0.13) | -0.19*** (-0.26,-0.12) | -0.20*** (-0.27,-0.13) | -0.20*** (-0.27,-0.13) | -0.20*** (-0.27,-0.13) | -0.20*** (-0.27,-0.13) |
| **Education** | | | | | | | | |
| College + | | — | — | — | — | — | — | — |
| Some College | | 0.18*** (0.09,0.27) | 0.18** (0.08,0.27) | 0.18** (0.08,0.27) | 0.18** (0.08,0.27) | 0.18** (0.09,0.27) | 0.18*** (0.09,0.27) | 0.18*** (0.09,0.27) |
| High School | | 0.23*** (0.13,0.32) | 0.22*** (0.13,0.32) | 0.23*** (0.14,0.32) | 0.23*** (0.13,0.32) | 0.23*** (0.13,0.32) | 0.23*** (0.13,0.32) | 0.23*** (0.14,0.32) |
| < High School | | 0.39*** (0.26,0.53) | 0.39*** (0.25,0.52) | 0.39*** (0.25,0.53) | 0.39*** (0.25,0.53) | 0.39*** (0.26,0.53) | 0.39*** (0.26,0.53) | 0.40*** (0.26,0.53) |
| **Quartile Wealth/Income** | | | | | | | | |
| 4 | | — | — | — | — | — | — | — |
| 3 | | 0.06 (-0.04,0.15) | 0.05 (-0.04,0.15) | 0.05 (-0.04,0.15) | 0.06 (-0.04,0.15) | 0.06 (-0.04,0.15) | 0.06 (-0.04,0.15) | 0.06 (-0.03,0.16) |
| 2 | | 0.20*** (0.09,0.30) | 0.17** (0.07,0.28) | 0.19** (0.08,0.29) | 0.19** (0.09,0.30) | 0.20*** (0.09,0.30) | 0.20*** (0.09,0.30) | 0.20*** (0.10,0.30) |
| 1 | | 0.32*** (0.21,0.44) | 0.29*** (0.18,0.41) | 0.31*** (0.19,0.42) | 0.32*** (0.20,0.43) | 0.32*** (0.21,0.44) | 0.32*** (0.21,0.44) | 0.32*** (0.21,0.44) |
| **Neighborhood Exposure** | | | 0.04 (0.00,0.08) | 0.03 (-0.01,0.07) | 0.01 (-0.03,0.05) | 0.00 (-0.01,0.02) | 0.00 (-0.01,0.01) | 0.00 (-0.01,0.01) |
| (Intercept) | -0.06** (-0.10,-0.03) | -0.22*** (-0.31,-0.14) | -0.20*** (-0.28,-0.11) | -0.22*** (-0.30,-0.14) | -0.22*** (-0.30,-0.14) | -0.25* (-0.44,-0.07) | -0.13 (-0.47,0.22) | -0.26*** (-0.37,-0.14) |
| R$^2$ | 0.012 | 0.059 | 0.060 | 0.060 | 0.059 | 0.059 | 0.059 | 0.059 |
| AIC | 8,973 | 8,843 | 8,841 | 8,842 | 8,844 | 8,845 | 8,844 | 8,844 |

Results of linear regression models with DPoAm aging as the outcome.

[1]ß (95% confidence interval)

*p<0.05

**p<0.01

***p<0.001

aging (Table 3, column 4). Neither physical disorder nor the air pollution measures are associated with either measure nor do they affect the association between DNAm aging and race.

To account for potential misclassification, we repeat these models excluding individuals who moved between 2010 and 2016 (S2 and S3 Tables). The overall magnitude of the race coefficient decreases across all models, but patterns in the magnitude of the racial disparity remain. The race coefficient is reduced upon addition of individual SES and SDI to the model. The association between SDI and GrimAge aging is attenuated but remains positive (β = 0.05, 95% CI 0.01, 0.09).

## Interaction models

To investigate differences in risk due to environmental exposures between racial groups, we implement linear regression models with interaction terms between each exposure measure

**Table 4. DPoAm aging: Interactions between environmental and social determinants and $PM_{2.5}$ pollution exposure.**

| DPoAm[1] | Race[1] | Gender[1] | Individual SES[1] | SDI[1] |
|---|---|---|---|---|
| **Race** | | | | |
| White | — | — | — | — |
| Black | -0.81 (-1.5,-0.09) | 0.21** (0.09,0.33) | 0.19** (0.07,0.32) | 0.16 (0.03,0.29) |
| **Gender** | | | | |
| Male | — | — | — | — |
| Female | -0.20*** (-0.27,-0.13) | 0.30 (-0.05,0.65) | -0.20*** (-0.27,-0.13) | -0.20*** (-0.27,-0.13) |
| **Education** | | | | |
| College + | — | — | — | — |
| Some College | 0.17** (0.08,0.27) | 0.18*** (0.09,0.27) | 0.18** (0.08,0.27) | 0.17** (0.08,0.27) |
| High School | 0.23*** (0.13,0.32) | 0.23*** (0.13,0.32) | 0.22*** (0.13,0.31) | 0.22*** (0.13,0.32) |
| < High School | 0.39*** (0.26,0.53) | 0.40*** (0.26,0.53) | 0.38*** (0.24,0.52) | 0.38*** (0.24,0.52) |
| **Quartile Wealth/Income** | | | | |
| 4 | — | — | — | — |
| 3 | 0.06 (-0.03,0.16) | 0.06 (-0.04,0.15) | -0.52 (-1.0,-0.07) | 0.06 (-0.04,0.15) |
| 2 | 0.20*** (0.09,0.30) | 0.20** (0.09,0.30) | -0.07 (-0.55,0.41) | 0.18** (0.08,0.29) |
| 1 | 0.32*** (0.20,0.43) | 0.33*** (0.21,0.44) | -0.30 (-0.79,0.19) | 0.30*** (0.18,0.42) |
| **Race * $PM_{2.5}$** | | | | |
| Black * $PM_{2.5}$ | 0.10* (0.03,0.17) | | | |
| **Gender * $PM_{2.5}$** | | | | |
| Female * $PM_{2.5}$ | | -0.05* (-0.09,-0.02) | | |
| **Quartile Wealth/Income * $PM_{2.5}$** | | | | |
| 3 * $PM_{2.5}$ | | | 0.06 (0.01,0.11) | |
| 2 * $PM_{2.5}$ | | | 0.03 (-0.02,0.08) | |
| 1 * $PM_{2.5}$ | | | 0.07 (0.02,0.12) | |
| **Social Deprivation Index** | | | | -0.14 (-0.32,0.05) |
| **Social Deprivation Index * $PM_{2.5}$** | | | | 0.02 (0.00,0.04) |
| **$PM_{2.5}$** | 0.00 (-0.02,0.01) | 0.03 (0.01,0.06) | -0.03 (-0.06,0.00) | 0.01 (-0.01,0.03) |
| **(Intercept)** | -0.19 (-0.38,0.00) | -0.54*** (-0.81,-0.27) | 0.08 (-0.22,0.39) | -0.27* (-0.46,-0.07) |
| $R^2$ | 0.061 | 0.062 | 0.062 | 0.061 |
| AIC | 8,839 | 8,838 | 8,841 | 8,842 |

Results of linear regression models with DPoAm aging as the outcome.

[1] ß (95% confidence interval)

*p<0.05

**p<0.01

***p<0.001

and race. Only the interaction between race and $PM_{2.5}$ with DPoAm aging as the outcome is statistically significant (Table 4 and Fig 2, full results in S4 Table). Black individuals appear more vulnerable to $PM_{2.5}$ exposure in terms of DPoAm aging (β = 0.10, 95% CI 0.03, 0.17; p<0.05). There were no significant interaction terms for GrimAge aging as the outcome (S5 Table).

Racial differences in risk from environmental exposures are due to structural and social determinants. To identify the determinants that may influence the relationship between $PM_{2.5}$ and DPoAm aging we implement models with DPoAm aging as the outcome and interaction terms between $PM_{2.5}$ and gender, education, wealth/income, SDI, and perceived social and physical disorder (S6 Table). There is a significant interaction between gender and $PM_{2.5}$,

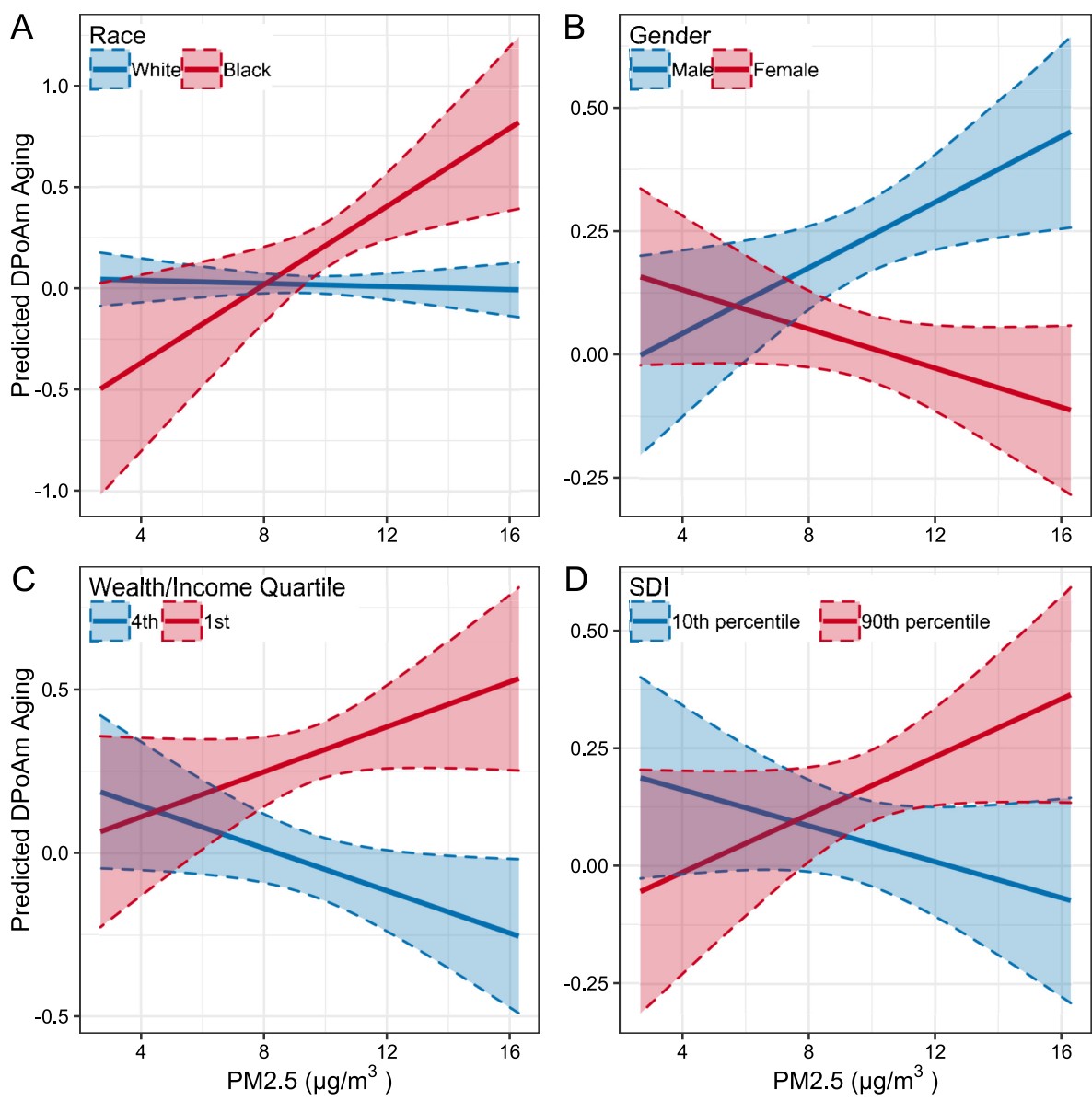

**Fig 2. Predicted association between DPoAm aging and PM_{2.5} by social determinants.** Estimated marginal mean values of DPoAm aging from models shown in Table 4. Interactions between PM_{2.5} and race (A), gender (B), wealth/income (C), and Social Deprivation Index (D) shown in solid lines, 95% confidence intervals shaded.

shown in Table 4 and Fig 2, where women appear less vulnerable to PM_{2.5} exposure (β = -0.05, 95% CI -0.09, -0.02; p<0.05). No other interaction terms are statistically significant; however, there are positive interactions with the lowest quartile of wealth/income (β = 0.07, 95% CI 0.12, 0.02) and with SDI (β = 0.02, 95% CI 0.0, 0.04). These interactions indicate that lower individual SES and higher neighborhood deprivation are associated with increased vulnerability to PM_{2.5} (Fig 2). Individual- and neighborhood-level SES may play a role in the relationship between PM_{2.5} exposure and DPoAm aging.

### Decomposition results

We use threefold Kitagawa-Blinder-Oaxaca decomposition to further quantify the contribution of individual and environmental variables to the racial gap in DNAm aging. Endowment terms quantify the contribution of different levels of the variables between groups and coefficient terms quantify the contribution of different relationships between the variables and DNAm aging between groups (Fig 3). Full results, including interaction terms, are shown in S7 Table. The negative endowment term of gender (GrimAge -0.08, 95% CI -0.12, -0.05; DPoAm -0.02, 95% CI -0.04, -0.01) indicates that if the gender balance were the same for both Black and White groups, the gap in DNAm aging would be larger. In this sample there are a greater portion of female Black participants (68%) than female White participants (57%), and female gender is associated with decreased DNAm aging. If the Black sample were also 57%

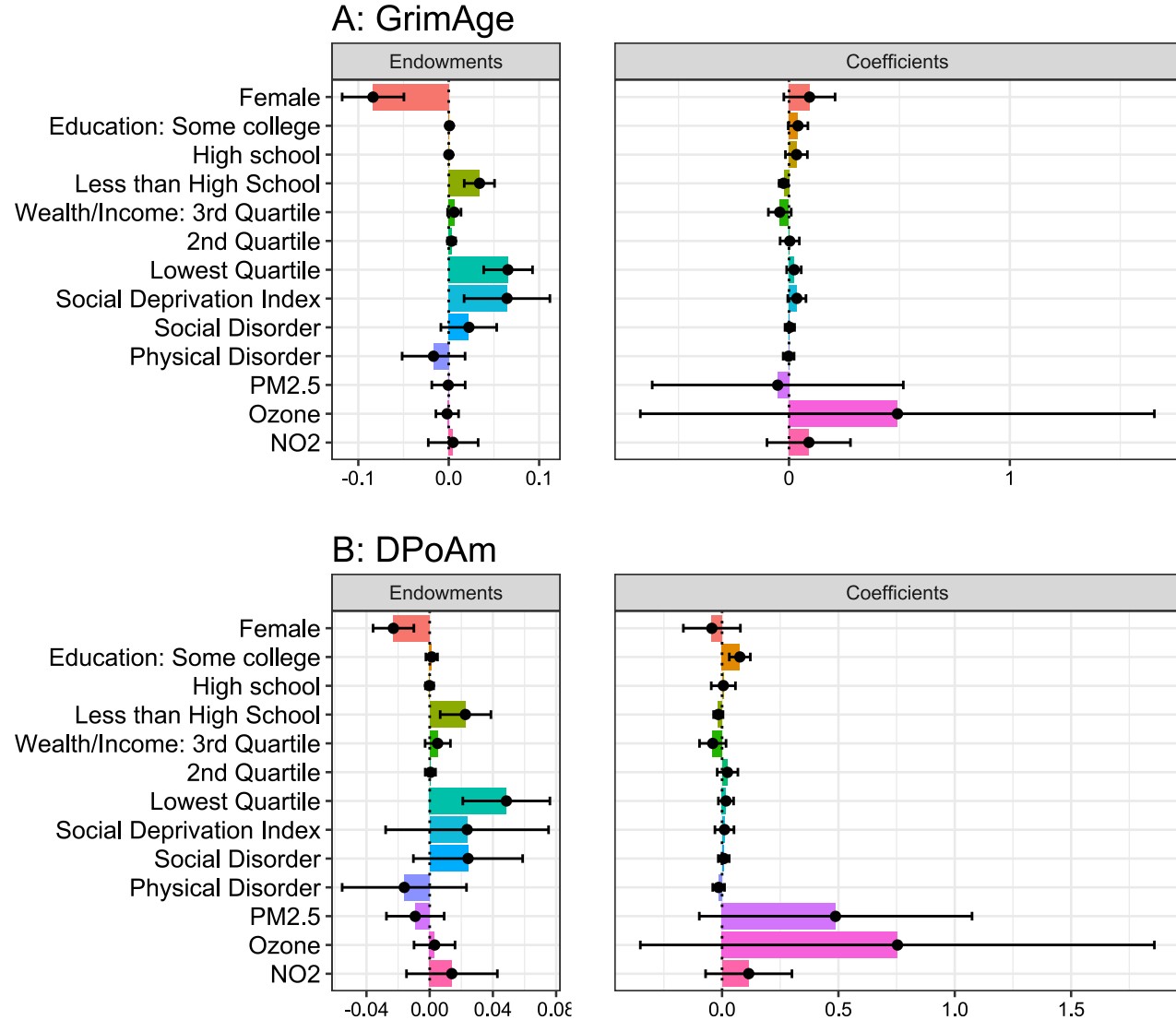

**Fig 3. Threefold decomposition of individual and neighborhood contributions to racial disparity in DNAm aging.** Magnitude of the endowment and coefficient terms and 95% confidence intervals for GrimAge (A) and DPoAm (B) aging.

female, their average DNAm aging would be even higher and the racial disparity would be larger (27.4% larger for GrimAge and 6.8% for DPoAm).

Differences in individual-level SES between racial groups contribute strongly to the race gap in DNAm aging. A positive coefficient term for some college for DPoAm (0.08, 95% CI 0.03, 0.12) shows that the relationship between receiving this level of education and DNAm aging may be different for Black and White individuals. A significant positive endowment term for less than high school (GrimAge 0.03, 95% CI 0.02, 0.05; DPoAm 0.02, 95% CI 0.01, 0.04) shows that more Black than White individuals receiving less than a high school education contributes to the gap in DNAm aging (11.1% for GrimAge, 6.7% for DPoAm). Similarly, a positive endowment term for the lowest quartile of wealth/income (GrimAge 0.07, 95% CI 0.04, 0.09; DPoAm 0.05, 95% CI 0.02, 0.08) shows that more Black individuals being at the lowest wealth/income level contributes to their higher levels of DNAm aging than White individuals (GrimAge 21.5%; DPoAm 14.4%).

The significant positive endowment term of SDI for GrimAge aging (0.06, 95% CI 0.02, 0.11) supports the linear regression results. Greater levels of neighborhood socioeconomic deprivation for Black participants contributes to the disparity in GrimAge aging (21.1%). The endowment term of SDI for DPoAm aging is positive but not statistically significant (0.02, 95% CI -0.03, 0.08). No other environmental exposures have significant endowment terms. The large coefficient term of $PM_{2.5}$ for DPoAm aging (0.49, 95% CI -0.09, 1.07) is not statistically significant but is consistent with the interaction model result that there may be racial differences in risk from $PM_{2.5}$ exposure which contribute to the disparity in DPoAm aging.

## Discussion

We investigated how individual- and neighborhood-level social determinants of aging contribute to weathering, measured by GrimAge and Dunedin Pace of Aging methylation (DPoAm) aging. We found that, as expected based on the literature, Black participants had faster DNA methylation (DNAm) aging and greater disadvantage in individual socioeconomic status (SES), neighborhood deprivation, perceived neighborhood disorder, and air pollution exposure than White participants. Lower levels of education and wealth/income for Black participants contribute substantially to the disparities in both GrimAge and DPoAm aging but did not fully explain them. Higher levels of neighborhood disadvantage for Black participants further contribute to the disparity in GrimAge, while greater risk due to fine particulate matter ($PM_{2.5}$) air pollution may contribute to the disparity in DPoAm. These findings suggest avenues for further research and action to advance progress toward eliminating racial disparities in aging.

While these results are largely consistent with previous literature, this work also presents novel findings. Work in social epigenetics has previously observed associations between DNAm aging with individual-level SES and neighborhood disadvantage, but to our knowledge this is the first study to quantify the contribution of these exposures to disparities in DNAm aging [7, 8, 21, 62, 70]. Inequitable levels of education, income, and wealth are well established as drivers of racial disparities in health; our findings from regression and decomposition analyses reinforce this body of evidence by showing that racial disparities in the level of education and wealth/income are the largest contributors to the disparity in DNAm aging [45]. Inequitable distribution between the highest and lowest levels of education and wealth/income between Black and White participants explains up to 33% of the disparity in GrimAge aging and 21% of the disparity in DPoAm aging.

This work also builds on the growing literature showing that disparities in neighborhood deprivation also contribute to racial health disparities. Associations between both GrimAge

and DPoAm aging and neighborhood deprivation have been found previously, however there have been mixed results on whether neighborhood deprivation has a significant independent effect after controlling for individual SES [7, 8, 21, 70]. We find a significant independent association and quantify the contribution of neighborhood deprivation to the racial disparity in GrimAge (21%). Perceived social and physical disorder do not contribute significantly to DNAm aging disparities, reflecting the less consistent evidence of associations with health for subjective measures of the environment compared to objective measures [25]. More work is needed to determine whether other aspects of the neighborhood, such as the built environment or crime rates, contribute to racial disparities in aging and to ascertain the mechanisms by which neighborhood deprivation is associated with DNAm aging. Potential mediators include health behaviors, social networks, psychosocial wellbeing and stress.

Although associations have been found between air pollution and other epigenetic clocks and with DNAm in epigenome-wide association studies, this is the first study to our knowledge to assess associations between air pollution and GrimAge and DPoAm aging [71–73]. We find that only Black participants have a significant association between $PM_{2.5}$ and DPoAm aging. There are no significant associations for White participants, for GrimAge, nor for other air pollutants (nitrogen dioxide ($NO_2$) nor ozone ($O_3$)). We expected that $O_3$ would not be significantly associated with DNAm aging nor contribute to disparities given that $O_3$ exposure is similar across racial groups (indeed slightly higher for White participants in our sample) and that there is little evidence of association between $O_3$ and DNAm. In contrast, $NO_2$ exposure has a large racial disparity and is associated with DNAm and many health outcomes [39, 40, 46]. The null findings for $NO_2$ may be partly due to data limitations; the most recent year available is 2010, while data for $PM_{2.5}$ and $O_3$ are available from 2014. When we repeat the model with $PM_{2.5}$ using 2010 exposure, there is no longer a significant interaction with race (S4 Table).

Higher risk of Alzheimer's disease and mortality from the same measured $PM_{2.5}$ exposure for Black than for White Americans has previously been documented [53, 55]. Our results suggest that Black adults' risk may be higher for DPoAm aging as well and that individual- and neighborhood-level SES may play a role in this disparity. It remains unclear whether the increased risk is due to measurement error, where Black individuals have higher personal exposure levels than White individuals who live in a census tract with the same average $PM_{2.5}$ level, or to factors that influence sensitivity to the effects of $PM_{2.5}$ exposure. Social and structural determinants play a role in both levels of personal $PM_{2.5}$ exposure and in the effect of $PM_{2.5}$. For example, individual- and neighborhood-level SES may influence time spent outdoors, time spent outside one's neighborhood, and levels of indoor air pollution. They also contribute to psychosocial and physiological stress which could weaken one's defenses against $PM_{2.5}$.

Results for GrimAge and DPoAm are largely consistent with some notable distinctions. While gender and education are significantly associated with and contribute to disparities in both measures, the strength and magnitude of these associations and contributions tend to be larger for GrimAge. While only GrimAge is significantly associated with neighborhood social deprivation, only DPoAm shows a difference in $PM_{2.5}$ risk. These differences indicate that the two measures of DNAm aging may capture slightly different underlying biological processes or aspects of aging. Smoking was included as a biomarker in the creation of GrimAge, so if tobacco smoking and $PM_{2.5}$ influence DNAm at overlapping CpG sites the effect of $PM_{2.5}$ may be masked [13]. DPoAm was constructed using data from a birth cohort in Dunedin, New Zealand, at ages 26–38 while GrimAge used data from the Framingham Heart Study Offspring cohort which has a wider geographic and age range (average age approximately 70 years) [14]. These different training sets may contribute to GrimAge being more sensitive to late-life social and environmental conditions.

While both GrimAge and DPoAm measures were trained on predominantly White cohorts and may measure biological aging less accurately in Black populations, we demonstrate that the racial disparity remains after correcting for ancestry-informative principal components, suggesting this observation is not driven by differences in genetic ancestry (S1 Table). An additional phenotypic DNAm age measure, PhenoAge, does not reflect weathering, perhaps due to less sensitivity to socioeconomic disadvantage, and was not included in our analyses [5, 21]. Epigenetic clocks are composite measures of dozens to hundreds of methylation sites, so it is not possible to determine whether specific sites or genes drive the observed racial disparity using this approach. An important future direction in this field is to build DNAm aging measures using data from more diverse study populations and to investigate whether methylation levels of specific sites, regions, or genes differ in their association with physiological aging outcomes between racial groups.

Our finding that women have decelerated DNAm aging compared to men is also consistent with previous literature [8, 20, 21, 62]. It is unknown whether this effect is attributable to sex, gender, or a combination thereof given gaps in current measurement practices [11]. Future research on the structural, social, and biological determinants of this sex/gender difference is needed. Research using an intersectional approach, which recognizes that multiply marginalized groups such as Black women face unique structural and social conditions, is also needed.

These results benefit from several strengths and careful consideration of weaknesses. We use two complementary analytical approaches, regression and decomposition, and find generally consistent results. This study is well powered with a large sample and accounts for potential sampling bias by using population weights. Complete case analysis inherently adjusts for missingness but likely underestimates the racial disparity since participants excluded for missing data are more likely to be Black and have higher DNAm aging. Survival bias may also result in underestimation of the disparity in DNAm aging.

This study has some limitations. We are not able to adjust for length of tenure at participants' residential location. A sensitivity analysis excluding participants who changed location between 2010 and 2016 finds that the magnitude of the racial disparity was lower and the association between GrimAge and Social Deprivation Index (SDI) is attenuated. Future studies could include individual-level factors that may mediate the effects of SDI on GrimAge, including health behaviors such as diet and physical activity which are influenced by one's neighborhood environment.

There is also potential misclassification of neighborhood exposure levels. An individual's exposure to neighborhood deprivation or air pollution may differ from the average level in their census tract, depending on where in the tract they live and how much time they spend in different locations. More granular geographic data may more accurately capture neighborhood exposures but is not available in this study. One's personal exposure to air pollution is also influenced by indoor air pollution which is not widely monitored. Studies have shown relatively high correlations between personal and ambient exposure of $PM_{2.5}$ but correlations are lower for $O_3$ and $NO_2$ [74–76]. We use the most recent year of air pollution data available prior to when the outcome was measured (2014 for $PM_{2.5}$ and $O_3$, 2010 for $NO_2$) because past residential locations are not available for all participants, so we are not able to accurately capture cumulative exposure. 2010 $NO_2$ data may not accurately capture more recent exposure, but the Pearson correlation between 2010 and 2014 exposure for $PM_{2.5}$ and $O_3$ are 0.82 and 0.75, respectively, indicating that neighborhood pollution levels are relatively consistent across temporal sampling. A major limitation in this study and in the field is lack of longitudinal DNAm data. Availability of DNAm outcomes at only one time point precludes analysis of trajectories of biological aging and assessment of causality. There are also limited data sets that integrate data on social and environmental exposures with markers of biological aging, which limits the potential for cross-validation of results. It

will be important for future studies to collect longitudinal data to investigate whether change in exposures result in change to DNAm aging and which periods of the life course are most important for weathering. Data on a greater variety of social and structural determinants on the individual and geographic level will also be important to investigate which exposures are most important and which factors moderate susceptibility to exposures.

Disparities in DNAm aging mediate significant portions of the racial disparities in a variety of age-related health outcomes [5]. Eliminating disparities in biological aging, or weathering, would greatly reduce the persistent and pervasive disparities in health between aging Black and White Americans. It is crucial to identify the factors contributing to weathering and to take action to address them. This study and others suggest that eliminating the racial gaps in education, income, and wealth would go a long way toward alleviating weathering but are not sufficient to eliminate it [45]. Interventions on the neighborhood level are also needed, as is attention to differences in risk from pollutants between populations.

## Supporting information

**S1 Table. Racial disparity in DNAm aging with weights or ancestry-informative principal components.** Mean DNAm aging is shown in unscaled units.
(PDF)

**S2 Table. GrimAge aging: Multivariable regression models excluding individuals who moved 2010–2016.** Results of linear regression models with GrimAge aging as the outcome excluding 667 participants whose residential census tract changed.
(PDF)

**S3 Table. DPoAm aging: Multivariable regression models excluding individuals who moved 2010–2016.** Results of linear regression models with DPoAm aging as the outcome excluding 667 participants whose residential census tract changed.
(PDF)

**S4 Table. DPoAm aging: Interactions between neighborhood exposures and race.** Results of linear regression models with DPoAm aging as the outcome.
(PDF)

**S5 Table. GrimAge aging: Interactions between neighborhood exposures and race.** Results of linear regression models with GrimAge aging as the outcome.
(PDF)

**S6 Table. DPoAm aging: Interactions between neighborhood and social determinants and PM$_{2.5}$ pollution exposure.** Results of linear regression models with DPoAm aging as the outcome.
(PDF)

**S7 Table. Threefold decomposition of individual and neighborhood contributions to racial disparity in DNAm aging.** Full results of decomposition for GrimAge and DPoAm.
(PDF)

## Acknowledgments

The authors thank Drs. Sharon Xie, Courtney Boen, and Ganga Bey for their feedback during analyses and drafting this manuscript. We used R packages emmeans() (version 1.8.0, https://cran.r-project.org/web/packages/emmeans/) and gtsummary (version 1.6.1, https://cran.r-project.org/web/packages/gtsummary/index.html) to produce figures and tables for publication.

## Author Contributions

**Conceptualization:** Isabel Yannatos, Shana Stites, Corey T. McMillan.

**Formal analysis:** Isabel Yannatos.

**Funding acquisition:** Rebecca T. Brown.

**Investigation:** Isabel Yannatos, Shana Stites.

**Methodology:** Isabel Yannatos.

**Project administration:** Rebecca T. Brown.

**Software:** Isabel Yannatos.

**Supervision:** Corey T. McMillan.

**Visualization:** Isabel Yannatos.

**Writing – original draft:** Isabel Yannatos.

**Writing – review & editing:** Isabel Yannatos, Shana Stites, Rebecca T. Brown, Corey T. McMillan.

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
