## [Decision Letter · Decision Letter 0]

30 Jan 2023

PONE-D-22-27535Contributions of neighborhood social environment and air pollution exposure to Black-White disparities in epigenetic agingPLOS ONE

Dear Dr. McMillan,

Thank you for submitting your manuscript to PLOS ONE. After careful consideration, we feel that it has merit but does not fully meet PLOS ONE’s publication criteria as it currently stands. Therefore, we invite you to submit a revised version of the manuscript that addresses the points raised during the review process.

We look forward to receiving your revised manuscript.

Kind regards,

Kuang-Hsi Chang

Academic Editor

PLOS ONE

Journal Requirements:

This work was supported by National Institutes of Health (R01-AG066152, R01-AG070885, P30-AG072979), Pennsylvania Department of Health (2019NF4100087335), and Penn Institute on Aging. 

However, funding information should not appear in the Acknowledgments section or other areas of your manuscript. We will only publish funding information present in the Funding Statement section of the online submission form. 

This work was supported by National Institute on Aging: R01-AG066152 (CM) , R01- AG070885 (RB), P30-AG072979 (CM). Additional support includes Pennsylvania Department of Health (2019NF4100087335; CM), and Penn Institute on Aging (CM).

National Institute on Aging: https://www.nia.nih.gov

Pennsylvania Department of Health: https://www.health.pa.gov/Pages/default.aspx Penn Institute on Aging: https://www.med.upenn.edu/aging/

Reviewers' comments:

Reviewer's Responses to Questions

**Comments to the Author**

1. Is the manuscript technically sound, and do the data support the conclusions?

Reviewer #1: Yes

Reviewer #2: Yes

Reviewer #3: Yes

2. Has the statistical analysis been performed appropriately and rigorously? 

Reviewer #1: Yes

Reviewer #2: Yes

Reviewer #3: Yes

3. Have the authors made all data underlying the findings in their manuscript fully available?

Reviewer #1: Yes

Reviewer #2: Yes

Reviewer #3: Yes

4. Is the manuscript presented in an intelligible fashion and written in standard English?

Reviewer #1: Yes

Reviewer #2: Yes

Reviewer #3: Yes

5. Review Comments to the Author

Reviewer #1: This study assessed the associations between racial disparities and aging with several environmental factors. Aging is defined by DNA methylation. The authors have an ambitious objective trying to unravel the very complex relationships between racial disparity and aging. The article is interesting and well-structured. I have some suggestions for the manuscript:

1. Line 70-41. Other epigenetic measures can be used to define DNA methylation and biological aging. Why did the authors select these two measures? The authors could elaborate advantages of the methodology and provide rationales.

2. Line 114-115. I am not familiar with the threefold decomposition, could the authors state more details about this approach? Why not use structural equation modeling (SEM) to investigate the causal effects of racial disparity on DNA methylation?

3. Because the relationships between racial disparity and DNA methylation with environmental factors are quite complex, the authors could use a directed acyclic diagram (DAG) to illustrate the relationships.

4. Line 181-182. These pollutants, PM2.5, NO2, and O3 are criteria pollutants, and the monitoring data are continuously provided by the US EPA. Their venous blood samples were collected in 2016. I suggest the authors can calculate the average concentrations of pollutants per census tract for 2016 by using the US EPA monitoring data. Using the average concentrations of NO2 for 2010 would result in exposure misclassification.

5. Line 203. As stated above, please provide a DAG to illustrate exposure and outcome with confounders or mediators. Some of the factors are mediators that are not appropriate to adjust in the same model.

6. Line 228-229. What are the factors that may influence the vulnerability of PM2.5? please list the factors.

7. For multiple linear regression, the authors need to use VIF to assess the multicollinearity problem.

8. Line 59 Please spell out the term when it first appears (here DNAm). The authors need to check all terms and abbreviations. Several terms are repeated in the text, for example, in Line 70, Dunedin Pace of Aging Methylation (DPoAm), Line 161-162 Dunedin Pace of Aging methylation (DPoAm) appeared again.

Reviewer #2: This manuscript investigated the contributions of neighborhood social environment and air pollution exposure on racial disparities. The authors have clearly described the introduction, material and method, results, and discussion. I have only a few questions and comments for them, as shown below.

1. On page 4, line 68, please cite a reference for the sentence.

2. On page 4, lines 69-70, please cite a reference for the sentence.

3. On page 5, line 89, please add an abbreviation for nitrogen dioxide.

4. On page 5, lines 87-91, the authors described the association between air pollutants exposure and age-related diseases. However, these studies did not analyze the association between ozone and age-related diseases. Why did authors hypothesize that ozone is also a risk factor for biological aging, and further analyzed the association between ozone and age-related diseases?

5. Please subscript to PM2.5 in the manuscript.

6. On page 9, lines 178-182, I have a big concern about how can the authors be sure that the air pollutant data from the EPA are representative for all subjects. Moreover, old people spent most of their time indoors, which means that indoor air pollutant concentrations may be an important contributor to exposure assessment. Please also describe the limitation of air pollution exposure assessment in the manuscript.

7. Eating habits are also important factor influencing human aging. Did the authors collect eating habits data from participants? If not, how to avoid the effect of eating habits on the association analysis?

Reviewer #3: This manuscript is of interest to show that Black participants are more vulnerable to age-related health disparities between older Black and White Americans. This manuscript is well-written and structured. Before this manuscript can be accepted for publication, please address the following comments.

1. Please describe DNA methylation and biological aging more detail in the section of materials and methods, for example: providing the representiative experimental output data of DNA methylation and biological aging.

2. What are the genes that have higher proportion of DNA methylation and biological aging in Black?

3. Do the authors look into the real health data of Black participants who have higher DNAm and DPoAm aging? The authors provied reference about Higher risk of Alzheimer’s disease and mortality from the same measured PM2.5 exposure for Black than for White Americans, but how about the overall health in Black participants who has higer DNAm and DPoAm aging? For example: any cancer or neurodegenerative disorders?

6. PLOS authors have the option to publish the peer review history of their article (what does this mean?). If published, this will include your full peer review and any attached files.

Reviewer #1: No

Reviewer #2: No

Reviewer #3: No

---

## [Author Response · Author response to Decision Letter 0]

22 Mar 2023

Editorial Response

We would like to thank the editors for your consideration of our manuscript and efforts in the review process. We provide point by point responses to all editorial and reviewer comments (italicized below) and highlight all major changes in the body of the manuscript. 

We first address the journal’s style requirements by including using sentence case headings and correct file names along with other minor edits throughout.

We additionally updated our funding statement:

“This work was supported by National Institute on Aging: R01-AG066152 (CM) , R01- AG070885 (RB), P30-AG072979 (CM), 1K23AG065442 (SS), 1K23AG065442-03S1 (SS). Additional support includes Pennsylvania Department of Health: 2019NF4100087335 (CM) and SS; Penn Institute on Aging (CM and SS); Alzheimer’s Association: AARF-17-528934 (SS); and the Research Centers Collaborative Network (RCCN) of the NIA: U24AG058556 (SS).

National Institute on Aging: https://www.nia.nih.gov

Pennsylvania Department of Health: https://www.health.pa.gov/Pages/default.aspx

Penn Institute on Aging: https://www.med.upenn.edu/aging/

Alzheimer’s Association: https://www.alz.org

Research Centers Collaborative Network (RCCN) of the NIA: https://www.rccn-aging.org

We revised the Acknowledgements section to remove all funding information previously include in error. It now states:

“The authors thank Drs. Sharon Xie, Courtney Boen, and Ganga Bey for their feedback during analyses and drafting this manuscript. We used R packages emmeans() (version 1.8.0, https://cran.r-project.org/web/packages/emmeans/) and gtsummary (version 1.6.1, https://cran.r-project.org/web/packages/gtsummary/index.html) to produce figures and tables for publication.”

Data Availability: Data used in this study are available through a third-party, the Health and Retirement Study. However, once a researcher has obtained access to the data, they can use our code available at github.com/pennbindlab to recreate the minimal data set. Description of the data set and source: The Health and Retirement Study (HRS) is sponsored by the National Institute on Aging (grant number NIA U01 AG009740) and is conducted by the University of Michigan. It is a longitudinal panel survey study that collects in-depth survey interviews and biological samples from a representative sample of Americans 50 and older. Study participants’ geographic location and Contextual Data Resource data are available only under special agreement because they contain sensitive and/or confidential information.

Verification of permission to use the data set: Restricted Data Agreement # 2021-084 was approved on October 15, 2021. Information to apply to gain access: Researchers must apply for access through HRS at the site https://hrs.isr.umich.edu/data-products/restricted-data. The application and use of data are free of charge. IRB approval is required. All questions related to HRS Restricted data should be sent to hrsrdaapplication@umich.edu. 

Finally, please note that we additionally added an author, Shana Stites, who provided substantial contributions in the revisions of this manuscript.

Reviewer #1

This study assessed the associations between racial disparities and aging with several environmental factors. Aging is defined by DNA methylation. The authors have an ambitious objective trying to unravel the very complex relationships between racial disparity and aging. The article is interesting and well-structured. I have some suggestions for the manuscript:

Thank you for your supportive comments and thoughtful suggestions that we feel by addressing have improved the quality of the manuscript.

1. Line 70-41. Other epigenetic measures can be used to define DNA methylation and biological aging. Why did the authors select these two measures? The authors could elaborate advantages of the methodology and provide rationales.

We agree that there are many potential epigenetic measures to consider when defining biological aging. Many of these “clocks”, referred to as first generation epigenetic clocks, were designed to predict chronological age using DNA methylation (DNAm) and a limitation of them is that they do not necessarily capture variation in biological aging. More recently, second generation clocks, like GrimAge and Dunedin Pace of Aging methylation (DPoAm) used in this manuscript, are defined using biomarkers and phenotypic characteristics that better capture the complexity of biological aging. In addition, because we are investigating racial disparities in epigenetic aging, we focused on measures that have previously demonstrated a significant racial disparity. To reflect this rationale we amended page 4, paragraph 2 (lines 72-77): 

“An advantage of these measures is they were trained to predict phenotypic age based on biomarkers and mortality rather than chronological age alone, which may more accurately reflect biological aging processes [12]. Both GrimAge and DPoAm demonstrate racial disparities, reflecting weathering, and strongly predict many age-related outcomes, including lung disease, cognitive decline, functional decline, and mortality [13–20].”

We also added a sentence on page 27, line 504-507:

“An additional phenotypic DNAm age measure, PhenoAge, does not reflect weathering, perhaps due to less sensitivity to socioeconomic disadvantage, and was not included in our analyses [5,21].”

2. Line 114-115. I am not familiar with the threefold decomposition, could the authors state more details about this approach? Why not use structural equation modeling (SEM) to investigate the causal effects of racial disparity on DNA methylation?

While SEM could be a valid and useful approach in this work, we chose threefold decomposition because it allows us to test our hypotheses in a straightforward way. We hypothesize that unequal levels of individual-level socioeconomic status and neighborhood exposures and different effects of PM2.5 both contribute to the racial disparity in DNAm aging. Threefold decomposition innovatively allows us to simultaneously evaluates contributions from different levels of explanatory variables between groups and different effects of these variables on the outcome. We added details about this approach in a paragraph on page 6:

“We selected an approach using threefold decomposition, a technique commonly used in econometrics and social sciences to investigate disparities between groups, because it allows us to evaluate how the both the distribution of an exposure and differences in risk contribute to disparities. This method decomposes a difference in an outcome between two groups into three components that are 1) explained by differences in the level of explanatory variables between the groups, 2) explained by differences in the effect of explanatory variables on the outcome between the groups, and 3) an unexplained portion [56]. This is accomplished by estimating group-specific regression models and using a counterfactual approach, where the change in the outcome disparity is evaluated after replacing the disadvantaged group’s covariate and coefficient values with those of the reference group.”

3. Because the relationships between racial disparity and DNA methylation with environmental factors are quite complex, the authors could use a directed acyclic diagram (DAG) to illustrate the relationships.

5. Line 203. As stated above, please provide a DAG to illustrate exposure and outcome with confounders or mediators. Some of the factors are mediators that are not appropriate to adjust in the same model.

Thank you for these suggestions to incorporate a DAG to illustrate our hypothesized and reported relationships. We added a figure (Fig 1) to provide a DAG showing the relationships between variables. The main exposure is race and the outcome is DNAm aging. When evaluating the association between neighborhood exposures and DNAm aging, it is important to control for individual-level SES as it influences both one’s neighborhood environment and has a direct effect on DNAm aging. This is reflected in added text on page 12, lines 248-252:

“Models are informed by a directed acyclic graph showing proposed relationships between all measures (Fig 1). Race influences education, income, wealth, neighborhood exposures, and DNAm aging due to various manifestations of structural racism. Age and gender are important confounders of these effects.”

4. Line 181-182. These pollutants, PM2.5, NO2, and O3 are criteria pollutants, and the monitoring data are continuously provided by the US EPA. Their venous blood samples were collected in 2016. I suggest the authors can calculate the average concentrations of pollutants per census tract for 2016 by using the US EPA monitoring data. Using the average concentrations of NO2 for 2010 would result in exposure misclassification.

We agree that using nitrogen dioxide (NO2) exposure data from 2010 is a weakness in this study. Unfortunately, even though monitoring data are continuously provided by the EPA, results of land use regression models are not available in HRS more recently than 2010. Land use regression modeling predicts NO2 levels at the census tract level using input data from fixed-site regulatory monitors, satellite estimates, and GIS-derived land-use data. HRS includes data in the Contextual Data Resource from land use regression performed by Bechle et al. from 2000-2010. EPA models fine particulate matter (PM2.5) and ozone (O3) levels using Fused Air Quality Surface Using Downscaling (FAQSD) and data are available from 2002-2016 in the HRS Contextual Data Resource. We use exposure data from 2014 to correspond with the most recent study wave prior to the outcome measurement in 2016. We see high correlation between 2010 and 2014 levels of PM2.5 and O3, suggesting that air pollution levels are relatively constant over time for the study participants. To address this weakness, we added text on page 29, lines 543-549:

“We use the most recent year of air pollution data available prior to when the outcome was measured (2014 for PM2.5 and O3, 2010 for NO2) because past residential locations are not available for all participants, so we are not able to accurately capture cumulative exposure. 2010 NO2 data may not accurately capture more recent exposure, but the Pearson correlation between 2010 and 2014 exposure for PM2.5 and O3 are 0.82 and 0.75, respectively, indicating that neighborhood pollution levels are relatively consistent across temporal sampling.”

6. Line 228-229. What are the factors that may influence the vulnerability of PM2.5? Please list the factors.

The factors we assess are gender, education, wealth/income, and aspects of the neighborhood social environment (social deprivation index, perceived social disorder, and perceived physical disorder). We clarified this on page 13, lines 268-270:

“We assess factors that may influence vulnerability to PM2.5 exposure by adding interaction terms between PM2.5 and gender, education, wealth/income, and neighborhood social environment measures (SDI, perceived social and physical disorder).”

7. For multiple linear regression, the authors need to use VIF to assess the multicollinearity problem.

We agree this is an important consideration and revised the Methods to clarify that we indeed assessed variance inflation factors and found low multicollinearity. We added this to page 13, line 262-263:

“Variance inflation factors for all models are ≤ 1.5, indicating low multicollinearity.”

8. Line 59 Please spell out the term when it first appears (here DNAm). The authors need to check all terms and abbreviations. Several terms are repeated in the text, for example, in Line 70, Dunedin Pace of Aging Methylation (DPoAm), Line 161-162 Dunedin Pace of Aging methylation (DPoAm) appeared again.

We thank the reviewer for their attention to detail. We made every effort to check all terms and abbreviations. We spell out each term the first time it appears in a section (Introduction, Methods, etc.) and use the abbreviation otherwise.

Reviewer #2: 

This manuscript investigated the contributions of neighborhood social environment and air pollution exposure on racial disparities. The authors have clearly described the introduction, material and method, results, and discussion. I have only a few questions and comments for them, as shown below.

Thank you for your positive comments and helpful suggestions that we feel have significantly improved the quality of our manuscript.

1. On page 4, line 68, please cite a reference for the sentence.

2. On page 4, lines 69-70, please cite a reference for the sentence.

We added references to key papers on DNA methylation aging (refs 10, 12-14).

3. On page 5, line 89, please add an abbreviation for nitrogen dioxide.

5. Please subscript to PM2.5 in the manuscript.

We thank the reviewer for their attention to detail. We checked all instances of nitrogen dioxide and NO2 in the text and made sure to spell out the term the first time it appears in a section and otherwise use the chemical formula. However, we chose to not subscript PM2.5 to be consistent with field standards because this abbreviation does not reflect a chemical formula and is the accepted common abbreviation for particulate matter with a diameter of ≤2.5µm. 

4. On page 5, lines 87-91, the authors described the association between air pollutants exposure and age-related diseases. However, these studies did not analyze the association between ozone and age-related diseases. Why did authors hypothesize that ozone is also a risk factor for biological aging, and further analyzed the association between ozone and age-related diseases?

This is an important point and we agree that we did not adequately address in the introduction. Many of the studies cited did analyze the effects of ozone exposure as well and found null or small effects on health outcomes. Therefore, we hypothesize that ozone is not related to DNAm aging and we include it as a negative control in analyses. We clarified this on page 5, line 102:

“Exposure to fine particulate matter (PM2.5) air pollution is an established risk factor for mortality and several age-related diseases and nitrogen dioxide (NO2) is associated with several of these outcomes as well, while findings for ozone (O3) are mixed and show weak to no associations with health outcomes [37–41].”

We also clarified in the hypothesis statement on page 7, lines 145-146:

“We further hypothesize that neighborhood exposures with a larger racial disparity and stronger association with biological aging contribute more to the DNAm aging disparity (specifically, neighborhood deprivation, PM2.5, and NO2 contribute more than neighborhood disorder and O3, which is included as a negative control).”

6. On page 9, lines 178-182, I have a big concern about how can the authors be sure that the air pollutant data from the EPA are representative for all subjects. Moreover, old people spent most of their time indoors, which means that indoor air pollutant concentrations may be an important contributor to exposure assessment. Please also describe the limitation of air pollution exposure assessment in the manuscript.

We agree with the reviewer that there are significant limitations in air pollution exposure assessment. A major limitation is the lack of data on indoor air pollution, as American adults spend only 7-8% of their time outdoors on average (Klepeis et al. 2001). We were not able to find reliable estimates of time spent outdoors in different age groups. However, studies using personal monitoring devices for air pollution exposure have shown relatively high correlation between personal and ambient levels of PM2.5 (Brauer M., 2010). Correlations are lower for O3 and NO2, which may partially explain the weaker associations between these pollutants and health outcomes in many studies using ambient exposures. We amended paragraph 2 on page 29 (lines 537-544) to better describe the limitations of air pollution exposure data:

“There is also potential misclassification of neighborhood exposure levels. An individual’s exposure to neighborhood deprivation or air pollution may differ from the average level in their census tract, depending on where in the tract they live and how much time they spend in different locations. More granular geographic data may more accurately capture neighborhood exposures but is not available in this study. One’s personal exposure to air pollution is also influenced by indoor air pollution which is not widely monitored. Studies have shown relatively high correlations between personal and ambient exposure of PM2.5 but correlations are lower for O3 and NO2 [74].”

7. Eating habits are also important factor influencing human aging. Did the authors collect eating habits data from participants? If not, how to avoid the effect of eating habits on the association analysis?

We agree that diet is an important factor in aging and that it may influence DNA methylation aging. HRS does not collect data on diet or eating habits. Diet and other health behaviors, such as physical activity and smoking, may be influenced by the neighborhood environment and would act as mediators in our proposed pathway. For example, neighborhood deprivation is associated with diet quality (Gilham et al., 2020) and eating habits, smoking, and physical activity are mediators between neighborhood SES and allostatic load (Robinette et al., 2016). It is not necessary to control for these health behaviors as confounders in our analyses, but testing the mediating pathways between SES, neighborhood environment, and DNAm aging is an important future direction of this work. In addition, other measures of biological aging, such as metabolomic age, may better capture aspects of aging related to metabolic processes in the lifespan and be more sensitive to diet (Robinson et al., 2020). We address this briefly on page 29, lines 534-536 by adding:

“Future studies could include individual-level factors that may mediate the effects of SDI on GrimAge, including health behaviors such as diet and physical activity which are influenced by one’s neighborhood environment.”

Reviewer #3: 

This manuscript is of interest to show that Black participants are more vulnerable to age-related health disparities between older Black and White Americans. This manuscript is well-written and structured. Before this manuscript can be accepted for publication, please address the following comments.

We appreciate your positive review of this manuscript and feel that by addressing your comments we have improved the quality of the manuscript.

1. Please describe DNA methylation and biological aging more detail in the section of materials and methods, for example: providing the representative experimental output data of DNA methylation and biological aging.

We thank the reviewer for this suggestion. The methylation assay outputs data from 866,091 individual methylation sites, a subset of which are used to calculate epigenetic clock values for each participant. While we are not able to display raw DNA methylation data because it is not yet publicly available from HRS, we added additional details on the epigenetic clock measures on page 9 lines (188-194):

“GrimAge was trained using data from the Framingham Heart Study Offspring Cohort. Elastic net regression was used to create DNAm proxies for plasma protein biomarkers and smoking pack years, then the DNAm proxies were combined to predict time to death. The resulting GrimAge value is transformed to units of years. DPoAm was trained using data from the birth cohort Dunedin Study. The rate of change in 18 biomarkers from ages 26 to 38 were combined in a Pace of Aging measure, then elastic net regression was used to select CpG sites that predict Pace of Aging.”

2. What are the genes that have higher proportion of DNA methylation and biological aging in Black?

This is an important point but one we are not able to address directly. HRS has not yet provided raw DNA methylation data, only the output of various epigenetic clock algorithms. Each clock is a composite value of many methylation sites (1030 sites for GrimAge, 46 for Dunedin Pace of Aging methylation). In addition, these clocks were built using multi-step analyses so it is not straightforward to assess how individual methylation sites, or their corresponding genes, relate to physiological or biological aging processes. Nevertheless, examining DNA methylation in more detail may reveal important mechanistic insights to age-related health and racial disparities. We added additional text to address this point on page 28, lines 512-517:

“Epigenetic clocks are composite measures of dozens to hundreds of methylation sites, so it is not possible to determine whether specific sites or genes drive the observed racial disparity using this approach. An important future direction in this field is to build DNAm aging measures using data from more diverse study populations and to investigate whether methylation levels of specific sites, regions, or genes differ in their association with physiological aging outcomes between racial groups.”

3. Do the authors look into the real health data of Black participants who have higher DNAm and DPoAm aging? The authors provided reference about Higher risk of Alzheimer’s disease and mortality from the same measured PM2.5 exposure for Black than for White Americans, but how about the overall health in Black participants who has higer DNAm and DPoAm aging? For example: any cancer or neurodegenerative disorders?

We thank the reviewer for pointing out this question that we did not fully address. The analysis that the reviewer suggests was performed by Graf, et al. (ref 10), who examined associations between DNAm aging and various age-related health outcomes among the same HRS participants our study is based on. In subgroup analyses, they found smaller effect sizes of DNAm aging for Black than for White participants, but found a similar result for chronological age. This implies that while DNAm aging may less accurately predict health outcomes for Black individuals, this is not due to differential precision of DNAm aging measures. They also examined racial disparities in these outcomes and the contribution of racial disparities in DNAm aging. While they did not examine cancer or neurodegenerative disorders, likely due to insufficient sample size and few cases, we are working on a manuscript that will examine associations between DNAm aging and cognitive outcomes. We added discussion of the previously published results on page 4 (lines 81-88):

“Prior evidence from the Health and Retirement Study, the cohort focus of this study, suggests that the hazard ratio of mortality is 2.32 and 1.71 per standard deviation increase in GrimAge and DPoAm acceleration, respectively [5]. Acceleration in GrimAge or DPoAm was further associated with racial disparities and an increased risk of prevalent and incident functional limitations and chronic conditions and poorer self-rated health as well as age-related outcomes: GrimAge and DPoAm mediated 13-92% of disparities in functional status and decline, self-rated health, and mortality [5].”

---

## [Decision Letter · Decision Letter 1]

2 May 2023

PONE-D-22-27535R1Contributions of neighborhood social environment and air pollution exposure to Black-White disparities in epigenetic agingPLOS ONE

Dear Dr. McMillan,

Thank you for submitting your manuscript to PLOS ONE. After careful consideration, we feel that it has merit but does not fully meet PLOS ONE’s publication criteria as it currently stands. Therefore, we invite you to submit a revised version of the manuscript that addresses the points raised during the review process.

We look forward to receiving your revised manuscript.

Kind regards,

Kuang-Hsi Chang

Academic Editor

PLOS ONE

Journal Requirements:

Reviewers' comments:

Reviewer's Responses to Questions

**Comments to the Author**

1. If the authors have adequately addressed your comments raised in a previous round of review and you feel that this manuscript is now acceptable for publication, you may indicate that here to bypass the “Comments to the Author” section, enter your conflict of interest statement in the “Confidential to Editor” section, and submit your "Accept" recommendation.

Reviewer #1: All comments have been addressed

Reviewer #2: (No Response)

Reviewer #3: All comments have been addressed

2. Is the manuscript technically sound, and do the data support the conclusions?

Reviewer #1: Yes

Reviewer #2: Yes

Reviewer #3: Yes

3. Has the statistical analysis been performed appropriately and rigorously? 

Reviewer #1: Yes

Reviewer #2: Yes

Reviewer #3: Yes

4. Have the authors made all data underlying the findings in their manuscript fully available?

Reviewer #1: No

Reviewer #2: Yes

Reviewer #3: Yes

5. Is the manuscript presented in an intelligible fashion and written in standard English?

Reviewer #1: Yes

Reviewer #2: Yes

Reviewer #3: Yes

6. Review Comments to the Author

Reviewer #1: The authors have addressed my comments appropriately, and the manuscript has been improved substantially. I have no further comments.

Reviewer #2: I have only two comments left:

1. please subscript the PM2.5, this is a standard fomrat;

2. i agreed with the response from the authors for Q. 6. However, please cite articles from the past 3 years, because the changes could occur in building characteristics and the association between indoor and outdoor air pollutants.

Reviewer #3: (No Response)

7. PLOS authors have the option to publish the peer review history of their article (what does this mean?). If published, this will include your full peer review and any attached files.

Reviewer #1: No

Reviewer #2: No

Reviewer #3: **Yes: **Yi-Chao Hsu

---

## [Author Response · Author response to Decision Letter 1]

16 May 2023

Response to Editor & Reviewers

Editorial Response

We would like to thank the editors for your consideration of our manuscript and efforts in the review process. We provide point by point responses to all editorial and reviewer comments (italicized below) and highlight all major changes in the body of the manuscript. We feel these minor changes further increase the quality of the manuscript and appreciate the reviewers’ further thoughtful suggestions.

Journal Requirements: All reviewers responded that all journal requirements have been adequately addressed with one discrepant suggestions about Data Availability. Two reviewers agree that we are in compliance with the PLOS Data policy but one reviewer suggested we were not in compliance. We re-iterate that we clearly outlined data availability in our prior submission which is the required language from the Health & Retirement Study:

Data used in this study are available through a third-party, the Health and Retirement Study. However, once a researcher has obtained access to the data, they can use our code available at github.com/pennbindlab to recreate the minimal data set. Description of the data set and source: The Health and Retirement Study (HRS) is sponsored by the National Institute on Aging (grant number NIA U01 AG009740) and is conducted by the University of Michigan. It is a longitudinal panel survey study that collects in-depth survey interviews and biological samples from a representative sample of Americans 50 and older. Study participants’ geographic location and Contextual Data Resource data are available only under special agreement because they contain sensitive and/or confidential information.

Verification of permission to use the data set: Restricted Data Agreement # 2021-084 was approved on October 15, 2021. Information to apply to gain access: Researchers must apply for access through HRS at the site https://hrs.isr.umich.edu/data-products/restricted-data. The application and use of data are free of charge. IRB approval is required. All questions related to HRS Restricted data should be sent to hrsrdaapplication@umich.edu.

Reviewers #1 and #3

Thank you for your supportive comments and thoughtful suggestions that we feel by addressing have improved the quality of the manuscript.

Reviewer #2:

We appreciate your attention to detail and have edited the manuscript to change the PM2.5 font and update some suggested references. Specifically:

1. please subscript the PM2.5, this is a standard format.

We have changed all instances of “PM2.5” to PM2.5. Given this minor change we do not highlight throughout.

2. I agreed with the response from the authors for Q. 6. However, please cite articles from the past 3 years, because the changes could occur in building characteristics and the association between indoor and outdoor air pollutants.

We agree with the reviewer that relationships between indoor and outdoor air pollution may change over time. Unfortunately there are technical challenges to accurately measure personal air pollution exposure and data are limited, especially studies that are representative across the United States. A recent review discusses the technological and methodological challenges of measuring personal air pollution exposure (Brokamp et al. 2019). A review and meta-analysis of studies on PM2.5 showed that in the United States the correlation between personal and ambient exposure is 0.6 but did not examine NO2 nor O3 (Boomhower et al., 2022). Another recent review and meta-analysis examined the absolute difference between personal ambient exposure and found greater differences and more variability for NO2 than for PM2.5 but did not examine O3 (Evangelopoulos et al., 2020). We were not able to identify more recent studies comparing personal and ambient exposure of O3 in the United States. We added the latter two citations to the manuscript (references 75 and 76) on page 29 (highlighted in yellow in the marked-up version).

---

## [Editor Report · Decision Letter 2]

30 May 2023

Contributions of neighborhood social environment and air pollution exposure to Black-White disparities in epigenetic aging

PONE-D-22-27535R2

Dear Dr. McMillan,

We’re pleased to inform you that your manuscript has been judged scientifically suitable for publication and will be formally accepted for publication once it meets all outstanding technical requirements.

Kind regards,

Kuang-Hsi Chang

Academic Editor

PLOS ONE
---

## [Editor Report · Acceptance letter]

12 Jun 2023

PONE-D-22-27535R2 

Contributions of neighborhood social environment and air pollution exposure to Black-White disparities in epigenetic aging 

Dear Dr. McMillan:

I'm pleased to inform you that your manuscript has been deemed suitable for publication in PLOS ONE. Congratulations! Your manuscript is now with our production department. 

Kind regards, 

on behalf of

Dr, Kuang-Hsi Chang 

Academic Editor

PLOS ONE